# CONVERGING AND STABILIZING GENERATIVE ADVERSARIAL IMITATION LEARNING

## ABSTRACT

Generative adversarial imitation learning (GAIL) is a powerful framework for model-free imitation learning. GAIL extracts a policy from expert demonstrations by training the parameterized policy to fool a discriminator for the state-action pairs generated by the learned policy and experts. However, the training process of GAIL has oscillating behaviors, which spoils its performance and efficiency. In this paper, we study the stability of GAIL from the perspective of control theory. We first formulate the training process of GAIL as a system of differential equations and formally prove that GAIL never approaches the desired equilibrium. We then leverage methodologies from control theory to design control functions that not only push GAIL to the desired equilibrium but also achieve asymptotic stability in theory. Motivated by the theoretical results, we propose a controlled GAIL algorithm with a modified learning objective for the discriminator. We evaluate our algorithm for MuJoCo tasks. While the vanilla GAIL is unstable and cannot acquire the expert return on some tasks, our controlled GAIL can approach expert returns on all the tasks.

## 1  INTRODUCTION

Generative Adversarial Imitation Learning (GAIL) (Ho & Ermon, 2016) is an effective method to learn sequential decision-making policies such as controlling autonomous vehicles or robots (Choi et al., 2021; Bhattacharyya et al., 2022), modeling real-life agents (Song et al., 2018), and even text generation (Muthukumar et al., 2021). GAIL is an imitation learning (IL) method, which is a class of methods that includes behavioral cloning (BC) (Pomerleau, 1991; Esmaili et al., 1995) and inverse reinforcement learning (IRL) (Ng et al., 2000; Abbeel & Ng, 2004; Hadfield-Menell et al., 2016). Despite its straightforward implementation, BC suffers from accumulating distribution shifts (Codevilla et al., 2019) as a supervised learning algorithm. On the other hand, IRL requires high computational costs for bi-level optimization since it searches for the optimal reward function in the outer loop and explicitly solves the RL subproblem in the inner loop. Even though Ziebart et al. (2008) employs the max entropy IRL to improve the efficiency of IRL, the optimal reward function of IRL may not be unique. To resolve the issues of IRL, GAIL combines IRL with generative adversarial networks (GANs) (Goodfellow et al., 2014) and solves the inner-loop RL problem via minimax optimization. Given expert trajectories $\tau_E$ sampled from expert policy $\pi_E$, GAIL alternatively trains a policy generator $\pi$ and a discriminator $D$, and ultimately, the policy generator approaches the expert policy, and the discriminator becomes indistinguishable for the expert policy and generated policy. Similar to GANs, the training process of GAIL can be formulated as a minimax optimization problem, where the policy generator and the discriminator compete with each other in order to reach equilibrium.

Unfortunately, GAIL inherits the notorious and well-known issue of GANs–unstable in training (Mescheder et al., 2018; Luo et al., 2023b). While some theoretical work suggests that GAIL can attain global convergence under linear settings (Zhang et al., 2020; Guan et al., 2021), empirically, it is well observed that GAIL has oscillating training curves and is unable to approach expert policy. In this paper, we formally identify the issue that GAIL exhibits unstable training behaviors and cannot converge to the minimax equilibrium as desired by transforming GAIL into a dynamic system and proving its incompetence to reach optimal points. To address this matter, we design a controller to alter GAIL's training dynamic and push it to the desired equilibrium. We theoretically prove that

our controlled GAIL achieves *asymptotic stability* and illustrate the effectiveness of our controlled GAIL under the IMITATION framework (Gleave et al., 2022).

## 1.1 RELATED WORK

Previous works have been done to improve the performance of GAIL in terms of sample efficiency and robustness in different environments. For example, Kostrikov et al. (2018) uses a discriminator-actor-critic algorithm to improve the sample efficiency of GAIL and reduce the bias for the reward function. Fu et al. (2017) present adversarial inverse reinforcement learning (AIRL) that designs a different discriminator based on GAIL to represent the reward function of IRL so that its reward function is more robust to variations in environments. Wang et al. (2017) combine variational autoencoder (VAE) on demonstration trajectories to improve the robustness and avoid mode collapse.

**Convergence of GAIL** Some theoretical works are conducted for the stability analysis of GAIL. Syed et al. (2008) analyze the convergence and stability of apprenticeship learning. Chen et al. (2020) show that GAIL approaches a stationary point (may not be the optimal solution) with the gradient-based algorithm on general MDP and general reward function. Zhang et al. (2020) introduces a natural policy gradient (NPG) algorithm and claims to achieve sublinear convergence to the global optimal solution. However, this work has restrictive theoretical assumptions on linear MDP and linear reward function, which are not applicable to real-life environments. Guan et al. (2021) extend the convergence analysis to nonlinear MDP and nonlinear reward function. However, its convergence guarantee is under the assumption that the objective function is strongly concave. Both works of Zhang et al. (2020) and Guan et al. (2021) lack empirical evidence to show that GAIL converges as claimed.

**Stability Analysis with Control Theory** In GANs, many stabilizing methods are proposed with control theory. Xu et al. (2020) propose a linear controller and locally stabilizes GANs. Luo et al. (2023b) utilize Brownian motion controller and globally exponentially stabilize GANs.

## 2 MODELING GAIL AS A DYNAMIC SYSTEM

In this section, we transform the training process of GAIL into a system of differential equations named dynamic system. Given an action space $A$ and a state space $S$, a policy generator $\pi_\theta(a|s) \in \Pi$ and a discriminator function $D_\omega : S \times A \to (0, 1)$, GAIL alternatively updates between parameters $\theta$ and $\omega$. In this work, we instead directly consider the updates of $\pi$ and $D$ in their respective function spaces. We leverage the variational method to compute the differential equation from the objective function of the generator and discriminator to get the training dynamic of GAIL.

Notice that the objective function for GAIL is

$$\mathbb{E}_\pi[\log(D(s,a))] + \mathbb{E}_{\pi_E}[\log(1 - D(s,a))] - \lambda H(\pi), \tag{1}$$

where $D$ is the discriminator that takes a state-action pair and outputs the probability of this state-action pair from the expert policy. $\pi$ is the policy representing the generator, $\pi_E$ is the given expert policy, and $H(\pi) \equiv \mathbb{E}_\pi[-\log \pi(a|s)]$ is the casual entropy of policy $\pi$. Respectively, the objective functions for the discriminator and generator can be written as

$$\max_D V_1(D, \pi) = \mathbb{E}_\pi[\log D(s,a)] + \mathbb{E}_{\pi_E}[\log(1 - D(s,a))] \tag{2}$$

$$\min_\pi V_2(D, \pi) = \mathbb{E}_\pi[\log D(s,a)] - \lambda \mathbb{E}_\pi[-\log \pi(a|s)]. \tag{3}$$

Similarly to Luo et al. (2023a), we can use the variational method to take the derivative of the above objective functions with respect to time $t$ to get the differential equations representing the training dynamic of the discriminator and the generator. Let $D(s, a, t)$ and $\pi(a|s, t)$ represent the discriminator and generator networks, respectively, at training time $t$. We introduce time $t$ into this notation because both the generator and discriminator network vary with respect to the time $t$. Since $\pi_E$ is constant through the training dynamic, we intentionally omit the symbol $t$ for expert policy. The training dynamic of GAIL can be written as (detailed in appendix A.2)

$$\frac{dD(s,a,t)}{dt} = \frac{\partial V_1(D,\pi)}{\partial D} = \frac{\rho_{\pi_t}(s,a)}{D(s,a,t)} - \frac{\rho_{\pi_E}(s,a)}{1 - D(s,a,t)} \tag{4}$$

$$\frac{d\pi((a|s),t)}{dt} = \frac{\partial V_2(D,\pi)}{\partial \pi} = \rho_{\pi_t}(s)A^{\pi_t}(s,a), \tag{5}$$

where $\rho_{\pi_t}(s,a)$ is the occupancy measure of policy $\pi_t$, and $A^{\pi_t}$ is the advantage function, such that

$$\rho_{\pi_t}(s,a) := \pi(a|s,t)\rho_{\pi_t}(s) := \pi(a|s,t)\sum_{n=0}^{\infty} \gamma^n P(s_n = s|\pi_t)$$

$$\mathbb{E}_\pi[c(s,a)] = \int_s \int_a \rho_\pi(s,a)c(s,a)dads$$

$$Q^{\pi_t}(s,a) := \mathbb{E}_{\pi_t}[r(\bar{s},\bar{a}) + \lambda \log \pi_t(\bar{a}|\bar{s})|s_0 = s, a_0 = a]$$

$$A^{\pi_t}(s,a) := Q^{\pi_t}(s,a) - \mathbb{E}_s Q^{\pi_t}(s,a).$$

## 3    PRELIMINARY

In this section, we present definitions and theorems on stability analysis for a dynamic system in the form of Ordinary Differential Equations (ODE). We first define the concepts of equilibrium and stability for a dynamic system. Then, we present the related theorem in control theory for the criteria of a stable dynamic system.

### 3.1    CONVERGENCE OF A DYNAMIC SYSTEM

Suppose we have a dynamic system of ODE with initial value $x(0) = x_0$:

$$\frac{dx(t)}{dt} = f(x(t)) \tag{6}$$

**Definition 3.1. (Equilibrium)** (Ince, 1956) A point $\bar{x}$ is an *equilibrium* of system 6 if $f(\bar{x}) = 0$. Such an equilibrium is also called a *fixed point, critical point, steady state*.

When we are considering the convergence behavior of a dynamic system 6, we first need to make sure such an equilibrium exists, which is a necessary condition for a dynamic system to converge. *Remark* 3.2. A dynamic system is unable to converge if an equilibrium does not exist.

### 3.2    STABILIZING DYNAMIC SYSTEM WITH CONTROL THEORY

In control theory, a controller can be added into a dynamic system to alter performance. Given a dynamic system that does not converge, a controller can be designed to push the dynamic to the required equilibrium and boost the stability of this dynamic system.

**Definition 3.3. (Controller)** (Brogan, 1991) A *controller* to a dynamic system is a function $u(t)$ such that

$$\frac{dx(t)}{dt} = f(x(t)) + u(t) \tag{7}$$

Suppose $\{x(t)\}_{t\geq 0}$ is a solution of above system 7 with equilibrium $\bar{x}$. We define two types of stability: Lyapunov stability and asymptotic stability.

**Definition 3.4. (Lyapunov Stability)** (Glendinning, 1994) System 7 is *Lyapunov Stable* if given any $\epsilon > 0$, there exists a $\delta > 0$ such that whenever $\|x(0) - \bar{x}\| \leq \delta$, we have $\|x(t) - \bar{x}\| < \epsilon$ for $0 \leq t \leq \infty$.

**Definition 3.5. (Asymptotic Stability)** (Glendinning, 1994) System 7 is *asymptotic stable* if given any $\epsilon > 0$, there exists a $\delta > 0$ such that whenever $\|x(0) - \bar{x}\| \leq \delta$, we have $\lim_{t\to\infty} \|x(t) - \bar{x}\| = 0$.

*Remark* 3.6. A dynamic system can be Lyapnuov stable but not asymptotic stable. However, every asymptotic stable dynamic system is Lyapnuov stable.

To evaluate controller $u(t)$ in terms of stability, the following theorem establishes stability analysis with regard to dynamic system 7 on equilibrium $\bar{x}$.

**Theorem 3.7.** *(**Principle of Linearized Stability**) (La Salle, 1976) Given a dynamic system 7 with equilibrium $\bar{x}$, this system is asymptotically stable if all eigenvalues of $\mathbb{J}(f(\bar{x}) + u(t))$ have negative real parts, where $\mathbb{J}(f(\bar{x}) + u(t))$ represents the Jacobian of $f(x(t)) + u(t)$ evaluated at $\bar{x}$.*

**Corollary 3.8.** *If $\mathbb{J}(f(\bar{x}) + u(t))$ has positive determinate and negative trace, all eigenvalue of $\mathbb{J}(f(\bar{x}) + u(t))$ have negative real parts, therefore theorem 3.7 also holds.*

## 4 CONVERGENCE AND STABILITY OF GAIL

In this section, we start with the dynamic system of GAIL derived in section 2 and prove that GAIL is unable to converge the policy generator to the expert policy. We then further analyze the training dynamic for one step of update for a state-action pair and propose controllers for both the discriminator and the policy generator, which not only successfully converge to the desired equilibrium but also achieve asymptotic stability.

### 4.1 GAIL DOES NOT CONVERGE

Throughout the training process of GAIL, we expect our policy generator to estimate expert policy, and our discriminator becomes indistinguishable between the expert and generated policy. In this manner, We define the *goal functions* for the discriminator and the policy generator to be

$$D^*(t) = \frac{1}{2}, \pi^*(t) = \pi_E, \tag{8}$$

We substitute the goal functions of Eq. 8 for the discriminator and the policy generator to the dynamic system (Eq. 4 and Eq. 5) and get (detailed in appendix A.3)

$$\frac{dD^*(t)}{dt} = \frac{\rho_{\pi^*(t)}(s,a)}{D^*(s,a,t)} - \frac{\rho_{\pi_E}(s,a)}{1 - D^*(s,a,t)} = 0, \tag{9}$$

$$\frac{d\pi^*(t)}{dt} = \rho_{\pi_t^*}(s)A^{\pi_t^*}(s,a) \neq 0. \tag{10}$$

Since Eq. 10 does not equal 0, according to def. 3.1, our goal functions are not equilibrium points for GAIL's training dynamic. Therefore, as suggested in remark 3.2, GAIL is unable to converge to the goal functions, which means the policy generator cannot converge to the expert policy.

### 4.2 CONTROLLING THE TRAINING PROCESS OF GAIL

To further analyze the stability of GAIL, we zoom in on our objective function for policy within one step. Considering step $k$ of the trajectory with state $s_k$ and taking action $a_k$, and letting $p(s)$ be the probability of the state at $s$ on time step $k$, then the objective functions for the discriminator and the policy generator at step $k$ can be written as

$$V_D = \int_{a_k} \int_{s_k} p(s_k)\pi(a_k|s_k)\log D(s_k,a_k)dsda + \int_{a_k} \int_{s_k} p(s_k)\pi_E(a_k|s_k)\log(1 - D(s_k,a_k))dsda$$

$$V_\pi = \int_{a_k} \int_{s_k} p(s_k)\pi(a_k|s_k)\log D(s_k,a_k)dsda + \lambda \int_{a_k} \int_{s_k} p(s_k)\pi(a_k|s_k)\log(\pi(a_k|s_k))dsda \tag{11}$$

We compute the one-step training dynamic for policy generator by taking derivative with respect to function $D$ and policy generator $\pi$ to get

$$\frac{dD(t)}{dt} = \frac{\partial V_D}{\partial D} = \frac{p(s)\pi(a|s,t)}{D(s,a,t)} + \frac{p(s)\pi_E(a|s,t)}{D(s,a,t) - 1} \tag{12}$$

$$\frac{d\pi(t)}{dt} = \frac{\partial V_\pi}{\partial \pi} = p(s)(\log(D(s,a,t)) + \log(\pi(a|s),t) + 1), \tag{13}$$

where $p(s)$ is the probability of state $s$ at step $k$. We define $x(t) = D(t)$, $y(t) = \pi(t)$, $\pi_E = E$, $p(s) = c$, and rewrite our dynamic system as

$$\frac{dx(t)}{dt} = \frac{cy(t)}{x(t)} + \frac{cE}{x(t) - 1} \tag{14}$$

$$\frac{dy(t)}{dt} = c \log x(t) + c\lambda \log y(t) + c\lambda \tag{15}$$

The training dynamic of GAIL in Eq. 4 and Eq. 5 does not converge to the goal functions. Fortunately, in control theory, we can design controllers to push a dynamic system to given goal functions. For example, a linear negative feedback control (Boyd & Barratt, 1991) can be applied to a dynamic system to reduce the oscillation of the system. We introduce our controlled system as

$$\frac{\mathrm{d}x(t)}{\mathrm{d}t} = \frac{cy(t)}{x(t)} + \frac{cE}{x(t) - 1} + u_1(t) \tag{16}$$

$$\frac{\mathrm{d}y(t)}{\mathrm{d}t} = c \log x(t) + c\lambda \log y(t) + c\lambda + u_2(t), \tag{17}$$

where $u_1(t)$ and $u_2(t)$ are the controllers for the discriminator and policy generator respectively. Since the derivative of the discriminator with respect to time evaluated at the goal function in Eq. 9 already equals 0, the discriminator is possible to approach its goal function. Therefore, we only need to design a linear negative feedback controller $u_1(t)$ for the discriminator to keep Eq. 16 equal to 0 at the goal function. On the other hand, the derivative of the policy generator with respect to time evaluated at its goal function in Eq. 10 does not equal 0. Therefore, $u_2(t)$ should be able to adjust Eq. 17 to 0 evaluated at the goal function. Here we define $u_1(t)$ and $u_2(t)$ to be the following functions

$$u_1(t) = -k(x(t) - \frac{1}{2}) \tag{18}$$

$$u_2(t) = -c\lambda \log E - c \log \frac{1}{2} - c\lambda + \alpha \frac{y(t)}{E} - \alpha \tag{19}$$

where $k, \alpha$ are hyper-parameters introduced. Intuitively, as $k$ gets larger, the discriminator will get a higher punishment as it deviates from the optimal value, so the discriminator would converge at a faster speed but may also have a larger radius of oscillation. For a detailed convergence behavior analysis of controlled GAIL and specific bound on the range of $k$ and $\alpha$, we need to conduct stability analysis with respect to the controlled dynamic system.

### 4.3 ASYMPTOTICALLY STABLE OF CONTROLLED GAIL

In this section, we formally prove that our controlled training dynamic of GAIL in Eq. 16 and Eq. 17 is *asymptotically stable* and derive the bound of the relationship between $\lambda$ and $k$.

For simplicity, let us define $\boldsymbol{z}(t) = (x(t), y(t))^\top$, and a function $f$ such that

$$f(\boldsymbol{z}(t)) = \begin{pmatrix} \frac{cy(t)}{x(t)} + \frac{cE}{x(t)-1} - k(x(t) - \frac{1}{2}) \\ c \log x(t) + c\lambda \log y(t) - c\lambda \log E - c \log \frac{1}{2} + \alpha \frac{y(t)}{E} - \alpha \end{pmatrix} \tag{20}$$

Therefore, our controlled training dynamic of GAIL in Eq. 16 and Eq. 17 can be transformed to the following vector form

$$d(\boldsymbol{z}(t)) = f(\boldsymbol{z}(t))dt. \tag{21}$$

**Assumption 4.1.** We assume $\alpha, k \in \mathbb{R}, k > 0$, such that
(a)

$$ck\lambda + k\alpha < 0$$

(b)

$$-8c^2\lambda - 8c\alpha - 4c^2 - ck\lambda - k\alpha > 0$$

(c)

$$\frac{k^2 + 32c(c\lambda + \alpha)}{32c} < 0$$

**Theorem 4.2.** *Let assumption 4.1 holds. The training dynamic of GAIL in Eq.21 is **asymptotically stable**.*

*Proof.* To analyze the convergence and stability behavior of system 21, first we need to verify definition 3.1 to make sure our goal functions are equilibrium points. Then we apply theorem 3.7 to prove system 21 is asymptotically stable. Notice that $z^*(t) = (\frac{1}{2}, E)\top$, then we substitute this goal function to system 21

$$d(\boldsymbol{z}^*(t)) = f(\boldsymbol{z}^*(t)) = 0$$

We the compute the linearized system near the goal function such that

$$d(\boldsymbol{z}(t)) = \mathbb{J}f(z^*(t))z(t)dt, \tag{22}$$

where $\mathbb{J}$ is the Jacobian of function $f$. Therefore,

$$\mathbb{J}f(z^*(t)) = \begin{pmatrix} -\frac{cy(t)}{x(t)^2} - \frac{cE}{(x(t)-1)^2} - k & \frac{c}{x(t)} \\ \frac{c}{x(t)} & \frac{c\lambda}{y(t)} + \frac{\alpha}{E} \end{pmatrix}_{(\frac{1}{2}, E)} = \begin{pmatrix} -8cE - k & 2c \\ 2c & \frac{c\lambda+\alpha}{E} \end{pmatrix} \tag{23}$$

Then we compute the determinate and trace of $\mathbb{J}f(z^*(t))$, which

$$det(\mathbb{J}f(z^*(t))) = \frac{(-8c^2\lambda - 8c\alpha - 4c^2)E + (-ck\lambda - k\alpha)}{E} \tag{24}$$

$$trace(\mathbb{J}f(z^*(t))) = \frac{-8cE^2 - kE + c\lambda + \alpha}{E} \tag{25}$$

Since $E = \pi_E(a|s)$ has range $[0, 1]$, therefore we have $det(\mathbb{J}f(z^*(t))) > 0$, if

$$ck\lambda + k\alpha < 0 \tag{26}$$

$$-8c^2\lambda - 8c\alpha - 4c^2 - ck\lambda - k\alpha > 0 \tag{27}$$

The graph of $trace(\mathbb{J}f(z^*(t)))$ is also a downward hyperbola with middle point $(\frac{-k}{16c}, \frac{k^2+32c(c\lambda+\alpha)}{32c})$. Therefore, $trace(\mathbb{J}f(z^*(t))) < 0$, if

$$\frac{k^2 + 32c(c\lambda + \alpha)}{32c} < 0. \tag{28}$$

As a result, system 21 is asymptotically stable if assumptions 4.1 hold.

$\square$

## 4.4 FROM CONTROLLER TO LOSS FUNCTION

Our controllers are designed for GAIL's training dynamic. However, we are particularly interested in reflecting controllers in loss functions. To do so, we take the integration of the dynamic system to get the controlled loss functions for the discriminator and generator and get

$$V_1'(D, \pi) = V_1(D, \pi) - \frac{k}{2}D^2(s, a) + \frac{k}{2}D(s, a) \tag{29}$$

$$V_2'(D, \pi) = V_2(D, \pi) - (c\log\frac{1}{2} + c\lambda\log E - \alpha)\pi(a|s) + \frac{\alpha}{2E}\pi^2(a|s), \tag{30}$$

where the value of hyperparameters $k$ and $\alpha$ should be bounded by assumption 4.1. Now we are concerning reflecting the new loss function into our algorithm. In practice, we are unaware of the expert policy for the generator's controller, so in the evaluation section, we only include the controller for the discriminator. In original GAIL, we sample $\tau$ and $\tau_E$ from policy $\pi$ and $\pi_E$ respectively and update the discriminator with function $\hat{\mathbb{E}}_\tau[\log D(s, a)] + \hat{\mathbb{E}}_{\tau_E}[\log(1 - D(s, a))]$. As a result, we modify the discriminator update with our controller as shown in Algorithm 1 below.

---

**Algorithm 1** The algorithm for controlled GAIL

---

1: **Input:** Expert trajectory $\tau_E$ sampled from $\pi_E$, initial parameters $\theta_0$, and $\phi_0$ for generator and discriminator.
2: **while** True **do**
3:     Sample trajectory $\tau$ from $\pi_\theta$.
4:     Update discriminator parameter $\phi$ with gradient

$$\hat{\mathbb{E}}_\tau[\log D(s,a) - \frac{k}{2}D^2(s,a) + \frac{k}{2}D(s,a)] + \hat{\mathbb{E}}_{\tau_E}[\log(1 - D(s,a)) - \frac{k}{2}D^2(s,a) + \frac{k}{2}D(s,a)] \quad (31)$$

5:     Update policy generator parameter $\theta$ as usual
6: **end while**

---

## 5 EXPERIMENT

In this section, we implement our designed controller in section 4.2 and evaluate its performance in different environments. Our experiments are based on the IMITATION framework (Gleave et al., 2022), which provides reliable baselines for imitation learning methods including BC, AIRL, GAIL, and dataset aggregation (DAgger) (Ross et al., 2011), simulated on various MuJoCo environments (Todorov et al., 2012). We evaluate and compare the performance of our controlled GAIL on Ant, Half Cheetah, Hopper, Swimmer, and Walker2d.

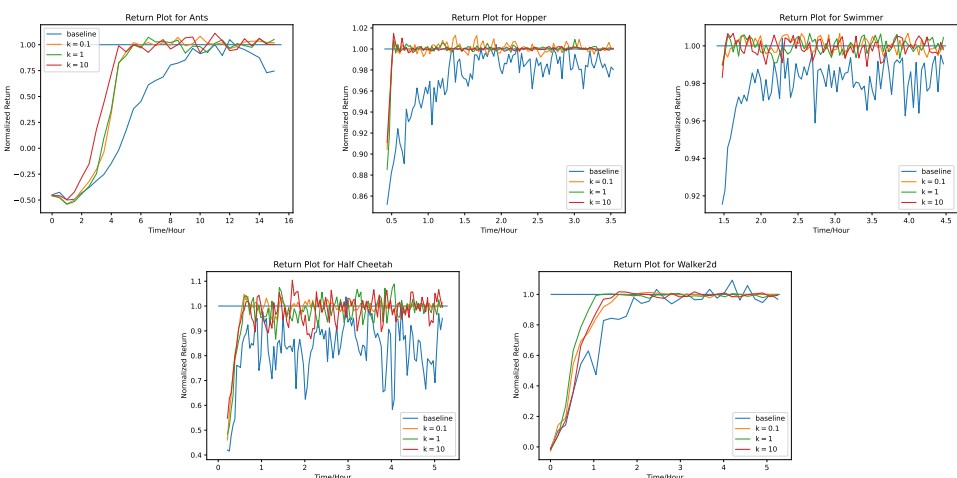

Figure 1: Normalized returns curves for controlled GAIL with $k = 0.1$, $k = 1$, and $k = 10$ on MuJoCo environments, where on the y-axis, 1 represents expert policy return and 0 represents random policy return

We replicate the exact experimental setup as reported in Gleave et al. (2022) for GAIL, BC, AIRL, and DAgger baselines. We modify the loss function of GAIL as in section 4.4 and evaluate our controlled GAIL with the same settings as in GAIL.

|  | Ant | Half Cheetah | Hopper | Swimmer | Walker2d |
|---|---|---|---|---|---|
| Random | $-349 \pm 31$ | $-293 \pm 36$ | $-53 \pm 62$ | $3 \pm 8$ | $-18 \pm 75$ |
| Expert | $2408 \pm 110$ | $3465 \pm 162$ | $2631 \pm 19$ | $298 \pm 1$ | $2631 \pm 112$ |
| Controlled GAIL | $2411 \pm 21$ | $3435 \pm 50$ | $2636 \pm 8$ | $298 \pm 0$ | $2633 \pm 12$ |
| GAIL | $2087 \pm 187$ | $3293 \pm 239$ | $2579 \pm 85$ | $295 \pm 3$ | $2589 \pm 121$ |
| BC | $1937 \pm 227$ | $3465 \pm 151$ | $2830 \pm 265$ | $298 \pm 1$ | $2672 \pm 95$ |
| AIRL | $-121 \pm 28$ | $1837 \pm 218$ | $2536 \pm 142$ | $269 \pm 8$ | $1329 \pm 134$ |
| DAgger | $3027 \pm 187$ | $1693 \pm 74$ | $2751 \pm 11$ | $344 \pm 2$ | $2174 \pm 132$ |

Table 1: Mean and standard deviation for returns of various IL algorithms and environments

As shown in figure 1, we evaluate and compare the performance of our controlled GAIL with naive GAIL for different hyperparameters of controllers with $k = 0.1$, $k = 1$, and $k = 10$. We normalize the return on the y-axis such that 0 refers to the random policy return and 1 refers to the expert policy return. The training curves of our controlled GAIL approaches to the expert policy at an earlier stage than GAIL and have a smaller range of osculations around the expert policy return.

We then compare our controlled GAIL with other IL methods, including BC, AIRL, and DAgger. From table 1, we can see that in each environment, the mean returns for controlled GAIL are closer to expert policy returns, and the standard deviations for controlled GAIL are smaller than other IL methods.

## 6   CONCLUSION AND DISCUSSION

In this work, 1) we formally establish the issue of unstable training for GAIL with control theory. 2) We formulate GAIL's training as a dynamic system and design a controller to stabilize and converge this dynamic system to the desired equilibrium. 3) Our controlled system achieves asymptotic stability in theory and successfully speeds up and stabilizes the training process of GAIL in experiments.

**Future Work** Even though our controller theoretically and empirically converges and stabilizes the return of GAIL to expert policy return, we are looking forward to maximizing return in reinforcement learning. Our controller may restrict the growth of generator policy return by forcing it to converge to expert policy return. In addition, when transforming GAIL's training to a dynamic system in section 2, we consider the training process of GAIL as a continuous dynamic system, whereas the alternative updating between the generator and the discriminator should be discrete in practice. Furthermore, we implement our controller only on the loss function of the discriminator for our evaluation since we are unaware of the expert policy included in the controller for the policy generator. However, future work can explore estimating the expert policy with numerical methods and add the controller for the policy generator as well.

**Border Impact** This work confronts the same social-ethical problem as other imitation learning methods, such as the potential invasion of privacy when collecting expert data and learning unwanted or unlawful behaviors from expert data. Additionally, when an agent performs an action that may be harmful, the legal responsibility is still unclear.

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

## A  APPENDIX

$$\max_D V_1(D, \pi) = \mathbb{E}_\pi[\log D(s,a)] + \mathbb{E}_{\pi_E}[\log(1 - D(s,a))] \tag{32}$$

$$\min_\pi V_2(D, \pi) = \mathbb{E}_\pi[\log D(s,a)] - \lambda \mathbb{E}_\pi[-\log \pi(a|s)]. \tag{33}$$

**Lemma A.1.** *Given that $\pi_\theta$ is a parameterized policy. Define the training objective for entropy-regularized policy optimization as*

$$J(\theta) = \mathbb{E}_{\pi_\theta}[r(s,a)] - \lambda \mathbb{E}_{\pi_\theta}[-\log \pi_\theta(a|s)].$$

*Its gradient satisfies*

$$\frac{\partial}{\partial \theta} J(\theta) = \mathbb{E}_{\pi_\theta}\left[\frac{\partial \log \pi_\theta(a|s)}{\partial \theta} Q^{\pi_\theta}(s,a)\right] = \mathbb{E}_{\pi_\theta}\left[\frac{\partial \log \pi_\theta(a|s)}{\partial \theta} A_E^{\pi_\theta}(s,a)\right],$$

*where $Q^{\pi_\theta}(s,a)$ and $A_E^{\pi_\theta}(s,a)$ are defined as*

$$Q^{\pi_\theta}(s,a) := E_{\pi_\theta}[r(\bar{s},\bar{a}) + \lambda \log \pi_\theta(\bar{a}|\bar{s})|s_0 = s, a_0 = a], \quad A^{\pi_\theta}(s,a) := Q^{\pi_\theta}(s,a) - \mathbb{E}_s Q^{\pi_\theta}(s,a).$$

*Proof.*

$$\frac{\partial}{\partial \theta} J(\theta) = \frac{\partial}{\partial \theta} \mathbb{E}_{\pi_\theta}[r(s,a)] - \lambda \mathbb{E}_{\pi_\theta}[-\log \pi_\theta(a|s)]$$

$$= \frac{\partial}{\partial \theta} \int \rho_{\pi_\theta}(s,a) r(s,a) \mathrm{d}a \mathrm{d}s + \lambda \frac{\partial}{\partial \theta} \int \rho_{\pi_\theta}(s,a) \log \pi_\theta(a|s) \mathrm{d}a \mathrm{d}s$$

$$= \int \frac{\partial \rho_{\pi_\theta}(s,a)}{\partial \theta} r(s,a) \mathrm{d}a \mathrm{d}s + \lambda \int \frac{\partial \rho_{\pi_\theta}(s,a)}{\partial \theta} \log \pi_\theta(a|s) \mathrm{d}a \mathrm{d}s + \lambda \int \rho_{\pi_\theta}(s,a) \frac{\partial \log \pi_\theta(a|s)}{\partial \theta} \mathrm{d}a \mathrm{d}s$$

$$= \int \frac{\partial \rho_{\pi_\theta}(s,a)}{\partial \theta}[r(s,a) + \lambda \log \pi_\theta(a|s)] \mathrm{d}a \mathrm{d}s + \lambda \int \rho_{\pi_\theta}(s) \pi_\theta(a|s) \frac{1}{\pi_\theta(a|s)} \frac{\partial \pi_\theta(a|s)}{\partial \theta} \mathrm{d}a \mathrm{d}s$$

$$= \int \frac{\partial \rho_{\pi_\theta}(s,a)}{\partial \theta}[r(s,a) + \lambda \log \pi_\theta(a|s)] \mathrm{d}a \mathrm{d}s + \lambda \int \rho_{\pi_\theta}(s) \frac{\partial}{\partial \theta} \int \pi_\theta(a|s) \mathrm{d}a \mathrm{d}s$$

$$= \int \frac{\partial \rho_{\pi_\theta}(s,a)}{\partial \theta}[r(s,a) + \lambda \log \pi_\theta(a|s)] \mathrm{d}a \mathrm{d}s$$

$$= \frac{\partial \mathbb{E}_{\pi_\theta}[r(s,a) + \lambda \log \pi_{\theta'}(a|s)]}{\partial \theta}\Big|_{\theta'=\theta}$$

The above derivation suggests that we can view the entropy term as an additional fixed reward $r'(s, a) = \lambda \log \pi_\theta(a|s)$. Applying the Policy Gradient Theorem, we have

$$\frac{\partial}{\partial \theta} J(\theta) = \mathbb{E}_{\pi_\theta}[\frac{\partial \log \pi_\theta(a|s)}{\partial \theta} Q^{\pi_\theta}(s, a)] = \mathbb{E}_{\pi_\theta}[\frac{\partial \log \pi_\theta(a|s)}{\partial \theta} A^{\pi_\theta}(s, a)],$$

where $Q^{\pi_\theta}$ is similar to the classic Q-function but with an extra "entrophy reward" term. $\square$

**Lemma A.2.** *The functional derivative for the optimization objective*

$$V(D, \pi) = \mathbb{E}_\pi[\log D(s, a)] - \lambda \mathbb{E}_\pi[-\log \pi(a|s)]$$

*satisfies*

$$\frac{\partial V}{\partial \pi} = \rho_\pi(s) A^\pi(s, a).$$

*where $A^\pi$ follows the same definition as in Lemma A.1.*

$$Q^\pi(s, a) := E_\pi[\log D(\bar{s}, \bar{a}) + \lambda \log \pi(\bar{a}|\bar{s})|s_0 = s, a_0 = a], \quad A^\pi(s, a) := Q^\pi(s, a) - \mathbb{E}_s Q^\pi(s, a).$$

*Proof.* Suppose $\pi$ is parameterized by $\theta$. The chain rule for functional derivative states

$$\frac{\partial V}{\partial \theta} = \int \frac{\partial V}{\partial \pi} \frac{\partial \pi}{\partial \theta} \mathrm{d}a\mathrm{d}s.$$

According to Lemma A.1, we have

$$\begin{aligned}\frac{\partial V}{\partial \theta} &= \mathbb{E}_\pi[\frac{\partial \log \pi(a|s)}{\partial \theta} A^\pi(s, a)] \\ &= \int \rho_\pi(s, a) \frac{\partial \log \pi(a|s)}{\partial \theta} A^\pi(s, a) \mathrm{d}a\mathrm{d}s \\ &= \int \rho_\pi(s) \frac{\partial \pi(a|s)}{\partial \theta} A^\pi(s, a) \mathrm{d}a\mathrm{d}s.\end{aligned}$$

Therefore, we have

$$\frac{\partial V}{\partial \pi} = \rho_\pi(s) A^\pi(s, a) = \rho_\pi(s)[Q^\pi(s, a) - \mathbb{E}_s Q^\pi(s, a)].$$

$\square$

**Proposition A.3.** *The constrained optimization problem*

$$\min_\pi V(D, \pi) = \mathbb{E}_\pi[\log D(s, a)] - \lambda \mathbb{E}_\pi[-\log \pi(a|s)] \quad s.t. \int \pi(a|s) = 1$$

*does not converge when $\pi = \pi_E$ and $D(s, a) = \frac{1}{2}$ for $\forall s, a$. Namely,*

$$\frac{\partial V}{\partial \pi}|_{\pi(s,a)=\pi_E(s,a), D(s,a)=\frac{1}{2}} \neq 0.$$

*When $\pi = \pi_E$ and $D(s, a) = \frac{1}{2}$, we have*

$$\begin{aligned}Q^\pi(s, a) &= E_{\pi_E}[\lambda \log \pi_E(\bar{a}|\bar{s}) - \log 2|s_0 = s, a_0 = a] \\ &= \sum_{n=0}^\infty \gamma^n \int p(s_n = \bar{s}|s_0 = s, a_0 = a) \int \pi_E(\bar{a}|\bar{s})[\lambda \log \pi_E(\bar{a}|\bar{s}) - \log 2]\mathrm{d}\bar{a}\mathrm{d}\bar{s} \\ &= -\sum_{n=0}^\infty \gamma^n \int p(s_n = \bar{s}|s_0 = s, a_0 = a)[\lambda H(\pi_E(\cdot|\bar{s})) + \log 2]\mathrm{d}\bar{s}\end{aligned}$$

$$\begin{aligned}A^\pi(s, a) &= Q^\pi(s, a) - \mathbb{E}_s Q^\pi(s, a) \\ &= \sum_{n=0}^\infty \gamma^n [p(s_n = \bar{s}|s_0 = s) - p(s_n = \bar{s}|s_0 = s, a_0 = a)]\lambda H(\pi_E(\cdot|\bar{s}))\end{aligned}$$

*According to Lemma A.2,*

$$\frac{\partial V}{\partial \pi} = \rho_{\pi_E}(s) A^{\pi_E}(s, a) = \rho_{\pi_E}(s) A^{\pi_E}(s, a)$$

*Apparently for different actions $a_1 \neq a_2$, we cannot guarantee $(s_n = \bar{s}|s_0 = s, a_0 = a_1) = (s_n = \bar{s}|s_0 = s, a_0 = a_2)$. Thus $\frac{\partial V}{\partial \pi}$ is not a constant and relies on action $a$. We have $\frac{\partial V}{\partial \pi}|_{\pi(s,a)=\pi_E(s,a), D(s,a)=\frac{1}{2}} \neq 0$.*

