# OpenReview forum: "Converging and Stabilizing Generative Adversarial Imitation Learning"
_ICLR.cc/2024/Conference — Submitted to ICLR 2024_

### Official Review · Reviewer_ruUB · 2023-10-22

**Soundness:** 2 fair
**Presentation:** 1 poor
**Contribution:** 1 poor
**Rating:** 3
**Confidence:** 4

**Summary:**

This paper claims that the original GAIL is not unstable from a control theory perspective. To address this issue, this paper introduces an additional controller in the optimization step. The paper conducts experiments to validate the proposed algorithm on MuJoCo locomotion control tasks.

**Strengths:**

This paper introduces control theory, such as stability analysis, into the imitation learning algorithm. This perspective is intriguing and may be further explored in future work.

**Weaknesses:**

- The definition of 'GAIL' is unclear

This paper primarily focuses on vanilla GAIL, which incorporates causal entropy into the objective function. However, in the introduction and related work sections, when discussing related research on GAIL, this paper appears to reference various GAIL variants. These other papers employ the state-action distribution matching principle as GAIL, utilizing different $f$-divergence measures. Unfortunately, this paper neglects to mention this aspect while reviewing the related work, causing confusion among readers and resulting in a misinterpretation of previous works.

- Motivation is unclear and week

This paper criticizes the stability of vanilla GAIL, i.e., the absence of a stationary point. This criticism is somewhat artificial or theoretical. This instability stems from the entropy regularization in the policy learning objective. Unless the expert policy is also the optimal policy with respect to the entropy-regularized MDP, the solution $D=1/2$ and $\pi= \pi^{E}$ is not a stationary point. Removing this regularized, GAIL is actually stable.

That being said, the key idea of GAIL, which is state-action distribution matching, is correct. So, why introduce this entropy regularizer in practice? The reason is the exploration issue when estimating the state-action (visit) distribution. In fact, there are more principled ways of accomplishing this [1, 2], although they involve slightly more complicated algorithm designs.

- Related Work is not enough

Many related topics are not reviewed:

The $f$-divergence perspective on state-action distribution matching (including GAIL) [3, 4, 5].

The sample complexity and generalization issues of imitation learning [6, 7, 8].

Note that researchers have successfully designed convergent state-action distribution matching methods under the tabular setting. One of the main gaps in practice is the generational gap that arises when using deep neural networks to approximate everything.



- Many lemmas and theorems have appeared in existing work

Lemmas A.1 and A.2 follow from the original GAIL work and the concept of entropy-regularized MDPs. This connection should be clarified.



- Experiment Results are Weak

Many strong baselines, such as the mentioned DAC and IQ-learn [9], are not compared. Thus, the significance of improvement of GAIL is unclear.

- Paper Writing Can be further improved

Many notations are not well-defined and explained. Please refer to the following comments.

1) The major concerns arise from the weak motivation and poor writing exhibited in this paper. The former has been discussed in the comments above, and I will now provide some comments and suggestions regarding the writing:

2) The formulation of the discounted MDP is not specified when modeling GAIL in Section 2.

3) Definition 3.1:
   "Such an equilibrium is also called a fixed point, critical point, or steady state."

4) Definition 3.5: The notation of $x_e$ is not defined.

5) Equations (9) and (10) may not hold true when the expert policy is the optimal policy with respect to the entropy-regularized MDP.

6) Section 4.2: "Let \(p(s)\) represent the probability of state \(s\) at time step \(k\)": this definition is very unclear. It seems that you are referring to the visitation probability."

7) The notation $\pi^E = E$ introduced in Equations (14) and (15) is confusing and unnecessary.

8) The hyper-parameter $k$ introduced in Equations (18) and (19) has been used before for other purposes.

9) The symbol $A^{\pi_{\theta}}_E$ is confused in Lemma A.1.

---

Reference:

[1]  Shani, Lior, Tom Zahavy, and Shie Mannor. "Online apprenticeship learning." *Proceedings of the AAAI Conference on Artificial Intelligence*. Vol. 36. No. 8. 2022.

[2] Liu, Zhihan, et al. "Provably efficient generative adversarial imitation learning for online and offline setting with linear function approximation." *arXiv preprint arXiv:2108.08765* (2021).

[3] Ghasemipour, Seyed Kamyar Seyed, Richard Zemel, and Shixiang Gu. "A divergence minimization perspective on imitation learning methods." *Conference on Robot Learning*. PMLR, 2020.

[4] Zhang, Xin, et al. "f-gail: Learning f-divergence for generative adversarial imitation learning." *Advances in neural information processing systems* 33 (2020): 12805-12815.

[5] Xu, Tian, Ziniu Li, and Yang Yu. "Error bounds of imitating policies and environments." *Advances in Neural Information Processing Systems* 33 (2020): 15737-15749.

[6] Rajaraman, Nived, et al. "Toward the fundamental limits of imitation learning." *Advances in Neural Information Processing Systems* 33 (2020): 2914-2924.

[7] Xu, Tian, et al. "Understanding Adversarial Imitation Learning in Small Sample Regime: A Stage-coupled Analysis." *arXiv preprint arXiv:2208.01899* (2022).

[8] Swamy, Gokul, et al. "Minimax optimal online imitation learning via replay estimation." *Advances in Neural Information Processing Systems* 35 (2022): 7077-7088.

[9] Garg, Divyansh, et al. "Iq-learn: Inverse soft-q learning for imitation." *Advances in Neural Information Processing Systems* 34 (2021): 4028-4039.

**Questions:**

- How to solve the inequalities in Assumption 4.1?
- Why choose the "time/hour" as the x-axis in Figure 1?
- How many expert trajectories are used in experiments ?
- Does this paper try any practical tricks when training GAIL? (refer to [10])



[10] Orsini, Manu, et al. "What matters for adversarial imitation learning?." *Advances in Neural Information Processing Systems* 34 (2021): 14656-14668.

---

> ### Author Response · Authors · 2023-11-22
> **Response for Review ruUB**
>
> Dear Reviewer ruUB,
>
> Thank you for your useful opinions.
>
> - We noticed that the unstable behavior of the naive GAIL comes from the entropy term. We are aiming to design a controller to stabilize the training with the entropy term in
> - Thank you for pointing out our need for more exploration in related work. We will include more aspects in our related work.
> - Yes, lemma A.1 and A.2 follow the original GAIL work; we use it to derive the training dynamic of GAIL
> - We finished the experiment on DAC on four expert trajectories with a faster convergence speed with our controller. We will include the result in our final submission
>
> Thank you for your comments on our notations; we will clarify them in our final submission. We are glad to address your questions.
>
> 1. We can use a graphic calculator to plot the feasible area of the inequality in assumption 4.1 concerning different values of c
> 2. Time/Hour in Figure 1 means the return is recorded concerning the training time, and the unit for the graph is per hour based. We can also plot the graph with the number of iterations being the x-axis. We did it on time to show that our controller did not increase the training time
> 3. The trajectory numbers are 32, 819000, 128, 32, 512 for Ant, HalfCheetah, Hopper, Swimmer, and Walker2d, respectively.

---

### Official Review · Reviewer_4PYc · 2023-10-29

**Soundness:** 2 fair
**Presentation:** 2 fair
**Contribution:** 3 good
**Rating:** 3
**Confidence:** 2

**Summary:**

The paper identifies that generative adversarial imitation learning (GAIL), casted as a differential equation, is generally an unstable dynamical system. Consequently, the paper proposes to use controllers to augment the differential equation such that it is stable. The paper provides theoretical justifications on the proposed controller, indicating that the augmented differential equation is stable. Finally, the paper provides experimental evaluation on the sensitivity of new hyperparameters induced by the proposed algorithm and compared the proposed algorithm against standard baselines in locomotion tasks.

**Strengths:**

- The paper demonstrates that the standard GAIL objective is unstable and leverages control theory to improve it.
- The experimental result seems to suggest that the introduced hyperparameter to be robust to varying values, and is able to recover expert performance.

**Weaknesses:**

**Comments**
- Regarding assumptions and clarity
	- The paper should indicate that the policy aims to minimize the cost. In particular, the problem setting is not well outlined---it should be said explicitly what assumptions we are making---Markov Decision Processes? Infinite horizon? Finite state-action spaces?
		- Since the problem formulation is not totally clear to me, I have a question regarding Eq. 10. If we assume the expert policy is deterministic with cost 0, then the advantage function is 0 correct? If so, in the case where the expert policy is deterministic with non-zero cost, can we not shift (presumably) the MDP's cost function?
	- Under section 4.2, is $p(s)$ not induced by a particular policy $\pi$? If so, will this $p$ be unstationary, thus Eq. 12 and Eq. 13 should be using a different $p$ at each $t$?
		- Furthermore, where did $s$ and $a$ come from? Should there still be an expectation?
	- It appears that the theory is an asymptotic result and assumes $D$ and $\pi$ realizability. I think the paper should clearly indicate these assumptions.
	- Under the proof of theorem 4.2
		- Between Eq. 25 and 26: Previously $E = \pi_E$ which is a probability distribution over action space $\mathcal{A}$ given state $s$. Why are we only considering a particular action in this case?
		- Also, what is the trace a function of, since $z^*$ is already set a priori?
- Regarding experimentation
	- Under section 4.4, in practice the paper only considers the controller for discriminator. Then the theory appears to not really matter since the practical algorithm does not address the stability of $\pi$? But, we should still expect the discriminator to converge faster with the controller?
	- The sensitivity analysis seems to be tested on a single seed---how confident are we in terms of replicating this result?
	- Since we can compare against the expert in experimentation, why is the paper unable to include the controller for the policy objective?

**Questions:**

- Definition 3.5: What is $x_e$?
- Figure 1: What does it mean by Time/Hour? What about the number of iterations?
- Table 1: Did we round to nearest integer? How many seeds? How many expert samples? Also why is DAgger degrading for both Half Cheetah and Walker?

**Possible typos**
- On page 1, introduction, line 5: "includes" instead of "include" and "behavioral" instead of "Behavioral"
- Section 2: "as a Dynamic" instead of "as Dynamic"
- Page 4, corollary 3.8: "eigenvalues" instead of "eigenvalue"
- Section 4, paragraph 1: "state-action" instead of "state action"
- Section 4.1, paragraph 2: Eq. 8 instead of Eq.8
- Section 4.1, third paragraph: def. 3.1 instead of def.3.1
- Page 4, section 4.2, line 3: $k$ instead of "k"
- Page 5, line 1: "state $s$ at step $k$"
- Page 5, paragraph 2, line 1: "described by Eq. 4 and Eq. 5"
- Page 5, two lines after Eq. 17: Eq. 9 instead of Eq.9
- Proof of theorem 4.2, Should $z^*(t) = (1/2, E)^\top$?

---

> ### Author Response · Authors · 2023-11-22
> **Response for Reviewer 4PYc**
>
> Dear Reviewer 4PYc,
>
> Thank you for your helpful thoughts. We respond to your questions below.
> - We assume our work under infinite horizon and infinite state-action space
> - $p(s)$ stands for the probability of our state at state $s$ at task time step $k$. The $t$ in Eq. 12 and Eq. 13 represents the training time.
> - The E in Eq. 25 and Eq. 26 is still the distribution function
> - The trace function is for the Jacobian of the dynamic system near the equilibrium point
> - In theory, we consider controlling both the discriminator and the generator, and in experiments, we only add the controller on the discriminator, and the entire training dynamic converges faster
> - The table we present is under three different seeds
> - $x_e$ is a typo and it should be $\bar{x}$
> - Time/Hour in Figure 1 means the return is recorded concerning the training time, and the unit for the graph is per hour based. We can also plot the graph with the number of iterations being the x-axis. We did it with respect to time to show that our controller did not increase the training time
>
> We also corrected those typos you have pointed out.

---

> > ### Comment · Reviewer_4PYc · 2023-11-22
> >
> > Thanks for the clarifications.
> > I have one major question:
> > 1. $p(s)$ needs to be dictated by some policy as it is a Markov Decision Process. What policy induces this distribution?

---

> > > ### Author Response · Authors · 2023-11-23
> > > **Reply for Reviewer 4PYc**
> > >
> > > In section 4.2, we treat $p(s_k)$ as a prior distribution since we are focusing on the agent time $k$ and analyze the dynamic of the objective function within one time-step. However, if we consider $p(s)$ for multiple steps, this distribution is induced by policy $\pi$.

---

### Official Review · Reviewer_S9nZ · 2023-10-31

**Soundness:** 3 good
**Presentation:** 2 fair
**Contribution:** 3 good
**Rating:** 3
**Confidence:** 4

**Summary:**

Generative Adversarial Imitation Learning (GAIL) is an effective technique for imitation learning, aiming to replicate expert behaviors. Despite its potential, GAIL's training often exhibits instability, akin to the challenges faced by Generative Adversarial Networks (GANs). This paper dives into GAIL's stability issues from a control theory perspective, framing its training dynamics as a system of differential equations and highlighting its inability to achieve equilibrium. By leveraging control theory, the authors redesign GAIL's training process to guide it towards the desired balance. When tested on MuJoCo tasks, this revised GAIL reliably matched expert performance, surpassing the original GAIL's inconsistent outcomes.

**Strengths:**

1. **In-depth Analysis**: The paper offers a thorough examination of GAIL's stability challenges, crucial for understanding the underlying reasons for its oscillating behaviors during training.

2. **Interesting Approach**: The researchers have employed control theory to address GAIL's instability, an interesting combination that bridges the gap between different areas of study.

3. **Theoretical Foundations**: The authors don't just propose a solution; they support their controlled GAIL method with rigorous theoretical proofs, enhancing the credibility of their approach.

**Weaknesses:**

1. **Limited Empirical Evidence**: Although the paper mentions evaluating their approach on MuJoCo tasks, the evaluation is basically based on reward from the learned policy. Considering that the proposed solution tries to control and drag the learned policy to the expert's, it might be more informative to show a comparison between the ground-truth policy and the learned one. In addition, the authors might want to add more experiments with varying hyperparameters, number of expert demonstrations, etc to make the experiment complete.

2. **Potential Assumptions**: It is unclear why in eq.~(30), the objective omits the parts except for $V_2(D,\pi)$. This implementation is inconsistent with Assumption 4.1, where $\alpha$ is a real number, and $c$ represents the state distribution. The authors did not shed light on this implementation.

3. **Writing & Notation**: The writing of this paper is not very clear. It would be better if the authors refine the notations and define them before using, for example, the $V^\prime$ in eq.~(29) and (30).

**Questions:**

1. Will the performance change with different $\alpha$ values?

2. Will adding the controller terms make the model more computationally costly to train?

3. Will the number of expert demonstrations change the performance?

---

> ### Author Response · Authors · 2023-11-21
> **Response for Reviewer S9nZ**
>
> Dear Reviewer S9nZ,
>
> Thank you for your valuable insights. We are happy to address your questions.
> 1. We conducted our experiments on GAIL-DAC, and our controller converges the training of GAIL-DAC faster. We will include our results in our final submission. We will also include more trials with different hyperparameters and various numbers of trajectories.
> 2. In our theoretical analysis, we design our controller for both the policy generator and the discriminator. However, as we mentioned in section 4.4, we are unaware of the expert policy, so in the evaluation section, we only add the controller for the discriminator.
> 3. Yes, we noticed that we have some typos and unclear notations in our submission. We will correct them in our final submission.
> 4. Our experiments are conducted with the controller only on the discriminator, so the $\alpha$ value is irrelevant here
> 5. The controller term does not make the model more computationally costly to train. Notice that Figure 1 is plotted with respect to time.
> 6. Our controller works with a different number of expert demonstrations. For example, the GAIL-DAC we include later is for only four expert demonstrations.

---

### Official Review · Reviewer_vkFG · 2023-11-05

**Soundness:** 1 poor
**Presentation:** 1 poor
**Contribution:** 1 poor
**Rating:** 3
**Confidence:** 4

**Summary:**

Motivated by a recent work on stabilizing GAN's training from the control-theoretic perspective (Luo et al., 2023a), the authors aimed to stabilize the training process in a way like (Luo et al., 2023a). Authors argued that GAIL's learning process does not converge to the equilibrium from control-theoretic perspective and suggested modified learning objectives having control functions (u functions) to address the issue. The proposed algorithm (named Controlled GAIL) was empirically evaluated with multiple MuJoCo tasks (which is a standard way of evaluating imitation learning algorithms these days), which tries to support Controlled GAIL's stability and converging to the expert scores.

**Strengths:**

- A control-theoretic approach to the training process of GAIL is interesting.
- Authors attempted to design a loss function with controller function based on a control theory.

**Weaknesses:**

- Ideas and derivations follow Luo 2023a. The proposed approach seems a straightforward extension of Luo 2023a to GAIL. I couldn't find out either theoretical or empirical challenges from the paper when Luo 2023a's idea (for GAN) was applied to GAIL.
- Empirical studies comparing Controlled GAIL and original GAIL should be improved. No hyperparameter ablation study was given, simply following the default setting of Gleave et al., 2022, which makes the empirical study in the paper unreliable.
- Readability and overall organization of writing need to be improved. Details will be mentioned below.

**Questions:**

- From the last line in page 2, authors referred Appendix A.2., as a derivation for the training dynamics of GAIL, but it seems only explaining Eq. (5) and does not support Eq. (4). Also, the step variable $t$ is missing in the proof, which makes it difficult to follow details.
- Most core ideas came from Luo 2023a. What is the novelty that differentiates this work from Luo 2023a?
- We normally consider a discrete step $t$, but the training dynamics considers derivatives w.r.t. $t$, which implies that $t$ is continuous. What is the motivation behind, and what is the rationale to use the continuous $t$?
- When indicating policies, the relation between $\pi_t$ and $\pi(a|s,t)$ is not rigorously mentioned, although they seem equal to each other.
- From my point of view, Section 3 should be mentioned before Section 2, so that we can see related work and preliminaries together, which will improve the readability.
- Remark 3.2.,  if it does not exist an equilibrium --> if an equilibrium does not exist.
- Definition 3.5., $x_e$ --> $\bar{x}$.
- Where do we use Theorem 3.7 and Corollary 3.8?
- Section 4.1's title (GAIL DOES NOT CONVERGE) is a strong argument. Since we can regard GAIL as a generalized version of GAN (e.g., GAIL becomes GAN in the contextual bandit setup), this argument implies GAN does not converge as well. (However, it seems that Luo 2023a didn't mention such statement in their paper.) Can you clarify this?
- In Section 4.2.'s Eq. (11), the same $p(s_k)$ is used for two expectations (one of the agent trajectory, the other for the expert trajectory), but I think the state distribution should be different for those two cases.
- From the last paragraph of 4.2., authors mentioned "..., so the discriminator would converge at a faster speed but may also have a larger radius of oscillation." Can you elaborate this statement?
- In Section 4.2., Assumption 4.1., what are the intuitive meanings for these assumptions? Also, are these assumption practical? For empirical studies, what are important factors? While $k$ was ablated in the experiments, how are other variables ($k$, $\alpha$) affecting the results?
- Do we have to show the full proof of Theorem 4.2. in the main part of the paper? Maybe a short intuitive proof sketch would be enough.
- In Section 4.4., authors mentioned "In practice, we are unaware of the expert policy for the generator’s controller, so in the evaluation section, we only include the controller for the discriminator." Would you elaborate this explanation?
- In Section 5. (Experiment), how many runs were considered? Would you share more details about Figure 1?

---

> ### Author Response · Authors · 2023-11-21
> **Thank you for your valuable comments**
>
> Dear Reviewer vkFG,
>
> Thank you for your valuable comments. We list our response below.
> - This work is inspired by Luo 2023a. However, GANs and GAIL have different training dynamics in terms of differential equations, so the BMC for GANs cannot be applied to GAIL. In our work, we design a new controller concerning the training dynamic of GAIL.
> - We also conduct our experiments on GAIL-DAC, and our controller converges the training of GAIL-DAC faster. We will include our results in our final submission.
> - Our proof in Appendix A.2 directly follows the original proof from the original GAIL. We did not include the one for Eq. 4 for discriminator because it is obvious (following the original paper)
> - The analysis is based on a continuous-time setting. However, in reality, the training dynamic should be discrete. We point out this difference in our discussion section
> - We use theorem 3.7 and corollary 3.8 to prove that our controlled dynamic system is asymptotically stable in theorem 4.2
> - Assumption 4.1 guides the relationship between the hyperparameters we introduced. The assumption is practical since we tried to plot it in a graphic calculator with different values of $c$. In experiments, we omitted $k, \alpha$.
> - Since the policy generator's equilibrium point is unknown, we cannot directly apply the controller for the generator to our experiment. Instead, we include the controller for the discriminator and achieve a faster rate of convergence in practice.
> - In section 5, figure 1 is generated by taking the average of 3 runs with respect to the training time, and table 1 is generated by taking the average of 3 seeds recorded at the end of training.
>
> We also thank you for pointing out our writing typos and ways to improve our readability; we will correct them in our final submission.

---

### Meta-Review · Area_Chair_i8Gs · 2023-12-12

**Metareview:**

This paper studies the convergence and stabilization of GAIL. Although this paper is interesting, as pointed out by all reviewers, most of the reviewers agree that the experiments are not sufficient to support the main claims, the novelty is somewhat limited, and the writing also needs to be improved. Based on my reading and the reviewers' comments, I recommend rejection.

**Justification For Why Not Higher Score:**

Limited novelty, inadequate experiments, and unclear writing.

**Justification For Why Not Lower Score:**

N/A

---

### Decision · Program_Chairs · 2024-01-16

Reject